# In Vitro Metabolism Study of Seongsanamide A in Human Liver Microsomes Using Non-Targeted Metabolomics and Feature-Based Molecular Networking

**DOI:** 10.3390/pharmaceutics13071031

**Published:** 2021-07-07

**Authors:** Zhexue Wu, Geum Jin Kim, So-Young Park, Jong Cheol Shon, Kwang-Hyeon Liu, Hyukjae Choi

**Affiliations:** 1Mass Spectrometry Based Convergence Research Institute, Kyungpook National University, Daegu 41566, Korea; wuzhexue@knu.ac.kr; 2College of Pharmacy and Research Institute of Cell Culture, Yeungnam University, Gyeongsan 38541, Korea; canta87@ynu.ac.kr; 3BK21 FOUR Community-Based Intelligent Novel Drug Discovery Education Unit, College of Pharmacy and Research Institute of Pharmaceutical Sciences, Kyungpook National University, Daegu 41566, Korea; soyoung561@hanmail.net (S.-Y.P.); sleier7640@naver.com (J.C.S.)

**Keywords:** *Bacillus* sp. KCTC 12796BP, metabolism, drug interaction, high-resolution mass spectrometry, seongsanamide A

## Abstract

Seongsanamide A is a bicyclic peptide with an isodityrosine residue discovered in *Bacillus safensis* KCTC 12796BP which exhibits anti-allergic activity in vitro and in vivo without significant cytotoxicity. The purpose of this study was to elucidate the in vitro metabolic pathway and potential for drug interactions of seongsanamide A in human liver microsomes using non-targeted metabolomics and feature-based molecular networking (FBMN) techniques. We identified four metabolites, and their structures were elucidated by interpretation of high-resolution tandem mass spectra. The primary metabolic pathway associated with seongsanamide A metabolism was hydroxylation and oxidative hydrolysis. A reaction phenotyping study was also performed using recombinant cytochrome P450 isoforms. CYP3A4 and CYP3A5 were identified as the major metabolic enzymes responsible for metabolite formation. Seongsanamide A did not inhibit the cytochrome P450 isoforms commonly involved in drug metabolism (IC_50_ > 10 µM). These results will contribute to further understanding the metabolism and drug interaction potential of various bicyclic peptides.

## 1. Introduction

Bicyclic peptides exhibit higher conformational rigidity, metabolic stability, and membrane permeability compared with linear or monocyclic peptides. Therefore, they are considered potential leads for the drug discovery stage [1,2]. The increased conformational rigidity of bicyclic peptides further improves their binding affinity to various drug targets and enhances target selectivity [1]. Several bicyclic peptide compounds have been reported from marine sponges (theonellamides) [3,4], microorganisms (thailandepsin B) [5], and mushrooms (amatoxin and phallotoxin) [6], which are used as antifungal agents, an HDAC inhibitor, and RNA polymerase inhibitors, respectively.

The genus *Bacillus* in the marine environment is known to produce structurally diverse peptides, including bicyclic peptides. Seongsanamide A (Figure 1) is one of four bioactive bicyclic depsipeptides isolated from *Bacillus safensis* KCTC 12796BP, which was obtained from a marine sponge collected from Seongsan on Jeju Island, Korea [7]. It contains a unique isodityrosine moiety which is responsible for its bioactivity [8]. This compound showed potent antiallergenic activities by suppressing leukotriene C4 production, and *β*-hexosaminidase release in bone marrow-derived mast cells without significant cytotoxicity, even at 100 µM [7]. In addition, it displays comparable anti-allergic properties with fexofenadine, a known H-1 blocker, after oral administration in a passive cutaneous anaphylaxis mouse model [7]. In 2020, Ye and coworkers reported the total synthesis of seongsanamide A using intramolecular Chan-Evans-Lam coupling and Miyuara borylation [8]. Although the biological activity and total synthetic method of seongsanamide A are known, its metabolism in human liver microsomes (HLMs) has not been described. The cyclic peptide daptomycin [9] is an antimicrobial agent that is primarily metabolized by hydrolysis and deacylation by actinomycetes [10] and glucagon-like peptide-1 amide. Cyclosporine is a prominent cyclic peptide drug that is metabolized by cytochrome P450 enzymes [11]. The linear polypeptide undergoes extensive N-terminal bond cleavage in mice and human hepatocytes [12]. However, the metabolism of bicyclic peptides, including seongsanamide A, has not been elucidated.

Recently, metabolomics has become a powerful tool for metabolite identification by combining mass spectrometry with multivariate statistical analysis. It has been used to identify the metabolic pathways of gefitinib [13], itraconazole [14], ketoconazole [15], and praziquantel [16]. In addition, feature-based molecular networking (FBMN) has also emerged as a powerful tool for metabolite identification enabling the organization and representation of tandem mass spectrometric data in a graphical form by mining similarities between MS/MS spectra along with their features, such as retention time [17,18,19]. Each molecular entity represents a precursor ion and its associated fragmentation spectrum. The links between the nodes indicate similarities between the mass spectra. By processing the tandem MS data, including alignment with its features, diverse isomers having identical chemical formulas may be distinguished in the molecular network. By propagating structural information within the network, the FBMN approach provides valuable insights into metabolite identification [20,21].

In this study, the in vitro metabolism of seongsanamide A was investigated in HLMs using both non-targeted metabolomics and FBMN. In addition, the chemical structure of seongsanamide A metabolites was identified by liquid chromatography-high resolution mass spectrometry (LC-HRMS). Cytochrome P450 (P450) isoforms responsible for seongsanamide A metabolism were characterized by a reaction phenotyping study using human recombinant P450 isoforms. The effects of seongsanamide A on the metabolism of nine P450 isoform-specific probe substrates were also evaluated to determine any potential drug interactions. The information will provide a better understanding of the pharmacokinetics and drug interactions of seongsanamide A.

## 2. Materials and Methods

### 2.1. Chemicals and Reagents

Seongsanamide A (>98%, Figure 1) was isolated from the fermentation broth extracts of *Bacillus* sp. KCTC 12796BP by as previously reported [7]. Acetaminophen, amodiaquine, bupropion, chlorzoxazone, dextromethorphan, trimipramine, uridine diphosphoglucuronic acid (UDPGA), nicotinamide adenine dinucleotide phosphate (NADP^+^), glucose-6-phosphate (G6P), and G6P dehydrogenase (G6PDH) were purchased from Sigma-Aldrich (St. Louis, MO, USA). Coumarin, midazolam, omeprazole, and tolbutamide were purchased from Toronto Research Chemicals (Toronto, ON, Canada). Pooled HLMs (XTreme 200) were supplied by XenoTech (Lenexa, KS, USA). We purchased rP450 isoforms (rCYP1A2, rCYP2A6, rCYP2B6, rCYP2C8, rCYP2C9, rCYP2C19, rCYP2D6, rCYP2E1, rCYP2J2, rCYP3A4, and rCYP3A5) from Corning life sciences (Woburn, MA, USA). All solvents used in the analyses were LC-MS grade (Fisher Scientific Co., Pittsburgh, PA, USA).

### 2.2. Incubation of Seongsanamide A in HLMs

Seongsanamide A (100 µM) was incubated with HLMs (1 mg/mL) or heat-inactivated HLMs (100 °C, 30 min) in 100 mM potassium phosphate buffer (pH 7.4) including an NADPH-generating system (3.3 mM G6P, 1.3 mM *β*-NADP^+^, 3.3 mM MgCl_2_, and 500 unit/mL G6PDH) for 60 min at 37 °C. The reaction was terminated by placing the tubes on ice and immediately adding 50 µL of acetonitrile. After centrifugation at 10,000× *g* for 5 min at 4 °C, aliquots of the supernatant were injected into a liquid chromatography-high resolution mass spectrometer (LC-HRMS).

### 2.3. Liquid Chromatography-High Resolution Mass Spectrometric Analysis

A Thermo Fisher UltiMate HPLC system coupled to a Q Exactive Focus orbitrap HRMS (Thermo Fisher Scientific Inc., Waltham, MA, USA) was used to identify seongsanamide A and its metabolites. Chromatography was performed on a Phenomenex Kinetex C18 column (100 × 2.1 mm, 2.6 µm, 100 Å). The mobile phase consisted of water containing 0.1% formic acid (A) and acetonitrile containing 0.1% formic acid (B) at a flow rate of 0.2 mL/min. Gradient elution was performed as follows: 10 to 95% B for 0–10 min, 95 to 10% B for 10–10.1 min, and 10% B for 10.1–15 min. A heated electrospray ionization source probe was used as an ion generator with nitrogen used as an auxiliary, sheath, and sweep gas. The mass spectrometry was operated in positive ionization mode for full scan MS^1^ with data-dependent MS^2^ mode. The full scan mode parameters were set as follows: spray voltage, 3.5 kV; capillary temperature, 320 °C; S-lens RF level, 50; auxiliary gas heater temperature, 350 °C; resolution, 70,000; scan rage, *m/z* 300–1800; and AGC target, 1 × 10^6^. The data-dependent spectra were obtained for the three most abundant peaks per cycle, and the parameters were set as follows: resolution, 17,500; normalized collision energy, 30 eV; and AGC target, 5 × 10^4^.

### 2.4. Metabolite Identification Using a Metabolomics Approach

The HRMS data were pre-processed with Compound Discoverer (version 2.1, Thermo Fisher Scientific), which was used to generate a data matrix containing information on *m/z* value, retention time, and peak intensity through peak alignment, isotope removal, and filtering. A multivariate analysis was performed using SIMCA (version 13, Umeå, Sweden). A principal component analysis (PCA) was used to identify significant differences between two groups (heat-treated HLMs vs. normal HLMs). The S-plots generated by the orthogonal partial least-squares-discriminant analysis (OPLS-DA) were used to identify potential metabolites by analyzing the ions contributing to the separation of two groups. A PCA score plot and an OPLS-DA loading S-plot were generated using Pareto scaling. The metabolite structures of seongsanamide A were identified according to the MS/MS fragmentation patterns compared with seongsanamide A.

### 2.5. Metabolite Identification Using a FBMN

The molecular network was created using data analysis workflow 2.0 on the Global Natural Products Social Molecular Networking web platform (GNPS, http://gnps.ucsd.edu accessed on 29 June 2021). The HRMS raw files were converted to an mzXML format, a text-based format used to represent mass spectrometry data describing the scan number, precursor *m/z* and the *m/z,* and intensity of each ion observed in MS/MS, using ProteoWizard 3.0.9935 [22,23]. The pre-processed data were exported to MZmine (version 2.53) and processed using feature detection and alignment [24]. The processes of mass detection at MS^1^ and MS^2^ levels in MZmine were conducted with noise levels of 1.0 × 10^3^ and 1.0 × 10^1^, respectively. The ADAP chromatogram builder algorithm was applied with a minimum group size of scans of 5, a minimum group intensity threshold of 5.0 × 10^3^, a minimum highest intensity of 1.0 × 10^3^, and an *m/z* tolerance of 5 ppm. Chromatogram deconvolution was used for additional processing by MZmine with the following conditions: a minimum peak height of 1.0 × 10^4^, a peak duration range of 0.05–10.00 min, a baseline level of 3.0 × 10^3^, an *m/z* range for MS^2^ scan pairing of 0.025 Da, and an RT range for MS^2^ scan pairing of 0.15 min. Isotopes grouping was applied with an *m/z* tolerance of 5.0 ppm, a retention time tolerance of 0.05 min (absolute), a maximum charge set at 3, and a representative isotope with the most intensity. The processed tandem mass data were aligned on the basis of their features, and a jointed peak list was made with the following conditions: an *m/z* tolerance of 5 ppm, a weight for *m/z* of 75, a retention time tolerance of 0.1 min (absolute), and a weight for RT of 25. The combined peak list with MS^2^ data was exported as a .mgf file and a quantitation table in csv format for molecular networking on the GNPS web platform. The parameters used to create a network of seongsanamide A and its metabolites were set as follows: a precursor ion mass tolerance of 0.02 Da and a fragment ion mass tolerance of 0.02 Da. The nodes were connected when the cosine score was greater than 0.8, and the MS/MS spectrum shared at least six matching peaks (https://gnps.ucsd.edu/ProteoSAFe/status.jsp?task=4e8fd03895a34397b859a8031b606e3d, accessed on 29 June 2021). The molecular network was visualized using Cytoscape (version 3.5.1) [25].

### 2.6. Reaction Phenotyping Study of Seongsanamide A Using Recombinant Cytochrome P450 Isoforms

The incubation mixture consisted of 100 mM potassium phosphate buffer (pH 7.4), 20 pmol/mL rCYP1A2, 2A6, 2B6, 2C8, 2C9, 2C19, 2D6, 2E1, 2J2, 3A4 or 3A5 isoforms, and seongsanamide A (1 µM) in a final volume of 100 µL. After a 5 min pre-incubation period at 37 °C, reactions were initiated by the addition of an NADPH generating system and further incubated for 15 min at 37 °C. The reaction was terminated by placing the incubation tubes on ice and by immediately adding 50 µL of acetonitrile. After centrifugation, aliquots of the supernatant were injected into an LC-tandem mass spectrometer (MS/MS). Seongsanamide A and its four metabolites (M1~M4) were separated on a Kinetex XB-C18 column (100 × 2.1 mm, Phenomenex) and analyzed using a Shimadzu LC-MS 8040 MS/MS (Shimadzu, Kyoto, Japan) equipped with a Nexera X2 LC system (Shimadzu) coupled with an electrospray ionization (ESI) interface. LC elution conditions for the analysis of seongsanamide A and its four metabolites were: 10 to 90% B for 0–6 min, 90 to 10% B for 6–6.1 min, and 10% B for 6.1–10 min. Detection of the ions was performed by monitoring the transitions of *m/z* 836.5 → 565.3 for hydrolyzed seongsanamide A (M1), *m/z* 1021.5 → 836.5 for monohydroxy-seongsanamide A (M2), *m/z* 1037.5 → 836.5 for dihydroxy-seongsanamide A (M3), and *m/z* 1053.5 → 836.5 for tridihydroxy-seongsanamide A (M4). Peak areas for all the metabolites were automatically integrated using LabSolutions software (Shimadzu).

### 2.7. Inhibitory Potency of Seongsanamide A against Human Cytochrome P450 Activity

The inhibitory potency of seongsanamide A was evaluated by a previously described method with slight modifications in the pooled HLMs [26,27]. The microsomal incubation was conducted using two cocktail sets containing P450-specific probe substrates; A: phenacetin (CYP1A2), bupropion (CYP2B6), amodiaquine (CYP2C8), tolbutamide (CYP2C9), omeprazole (CYP2C19), and dextromethorphan (CYP2D6); B: coumarin (CYP2A6), chlorzoxazone (CYP2E1), and midazolam (CYP3A). After incubation for 10 min at 37 °C in HLMs in the presence or absence of seongsanamide A (0, 0.5, 2, 5, 20, and 50 µM), the reaction was quenched with cold acetonitrile containing trimipramine as an internal standard. After centrifugation, aliquots of the supernatants were analyzed using LC-MS/MS as described previously [28]. IC_50_ (concentration of the inhibitor causing 50% inhibition of the original enzyme activity) values were calculated using WinNonlin (Pharsight, Mountain View, CA, USA).

## 3. Results and Discussion

To choose the optimum polarity mode, standard solutions of seongsanamide A were infused into the MS and analyzed in positive and negative ionization modes using an electrospray ionization source. The results showed that the chromatographic peaks in the positive mode were higher compared with those in the negative mode. Seongsanamide A is a cyclic depsipeptide with a peptide linkage, and it is easily protonated in a positive ionization mode. Therefore, the positive ionization mode was selected, which was consistent with previous studies [7].

### 3.1. Profiling of Seongsanamide A Metabolites Using a Metabolomic Approach

The metabolomics approach combined with a multivariate analysis has proven to be efficient for the unbiased identification of drug metabolites [14,15,16,29]. In this study, a metabolomics approach was applied to comprehensively identify seongsanamide A metabolites. The results of the multivariate statistical analysis for ions produced by the LC-HRMS analysis of HLM incubation samples, incubated in the presence or absence of heat pre-treatment for HLM, are shown in Figure 2. A PCA score plot revealed a clear difference between the normal HLM group and the heat-deactivated control group treated with seongsanamide A (Figure 2A). We obtained good-quality parameters using the PCA model (fitness (R^2^X) = 0.994; predictability (Q^2^) = 0.975), indicating that this model may be regarded as a predictable model [30]. The loading plot of the OPLS-DA (S-plot) was used to identify important variables contributing to group separation. The variables observed at the top right position in the S-plot were identified as metabolites of seongsanamide A [14,31], and they are marked on the S-plot (Figure 2B). The chemical structure of the four metabolites (M1~M4) of seongsanamide A was identified based on MS/MS data through the confirmation of the chemical formula for the key variables responsible for group separation (Table 1).

### 3.2. Profiling of Seongsanamide A Metabolites Using an FBMN

FBMN uses a vector-based computational algorithm to compare the degree of spectral similarity between each MS/MS spectra for a dataset that distinguishes each ion by retention time [19,32,33]. It has also been applied to the unbiased identification of metabolites [20,21,34,35]. In this study, an FBMN analysis was used to identify seongsanamide A metabolites. A molecular network was created by using the GNPS web platform for the HRMS raw data processed by MZmine [17,36]. Analysis of microsomal incubation samples allowed us to generate a multi-matrix molecular network displaying the MS/MS data acquired during analysis (Figure 3A). Nodes are represented by each ion with retention time, labeled with the *m/z* value, and identified metabolite name. A specific color was assigned to each group (red, heat-deactivated HLMs group; and blue, normal HLMs group). The colored area in each node represents the relative intensity of the corresponding variable (metabolite) under each condition. Nodes were linked together in a cluster according to the similarities of their MS/MS spectra. Visualization of the multi-matrix network shows a cluster containing seongsanamide A linked to other nodes (Figure 3B). In this cluster, five molecules were found to be related to seongsanamide A, and they are putative seongsanamide A metabolites. Among them, nodes having *m/z* 1021.56, 1037.56, and 1053.55 had the same *m/z* values as M2, M3, and M4, respectively, which were identified through the metabolomic approach. They were mono-, di-, and tri-hydroxyseongsanamide A. Metabolite M1 produced through peptide bond cleavage did not form a cluster with seongsanamide A because of low spectral similarity in the MS/MS spectra between seongsanamide A and M1.

### 3.3. Metabolite Structure Identification

The structures of the metabolites were determined based on the accurate mass and fragment ion patterns. The mass error was within 5 ppm. Representative chromatograms and MS/MS spectra of the metabolites are shown in Figure 4 and Figure 5. To identify the structure of the metabolites, we first determined the structural properties of seongsanamide A. The protonated molecular ion of seongsanamide A (P) was observed at *m/z* 1005.5647 (mass error < 1.0 ppm) and eluted at 7.8 min. The MS/MS spectrum of seongsanamide A by fragmenting *m/z* 1005.5647 through collision gave the base peak at *m/z* 836.4544 by amide bond cleavage (Simon et al. 2010) (Figure 4 and Figure 5A). Fragment ions were generated at *m/z* 808.4593 and 765.4174, indicating a loss of carbon monoxide (−28 Da) and the loss of an alanine residue (−71 Da) from the base peak. The neutral loss of carbon monoxide has been reported with cyclic peptides [37], benzodiazepine drugs [38], and flavonoids [39]. Fragment ions were also generated at *m/z* 977.5698 and 737.4225, indicating the loss of carbon monoxide (−28 Da) from fragment ions at *m/z* 1005.5647 and 765.4174, respectively (Appendix A).

The protonated molecular ion of metabolite M1 was observed at *m/z* 836.4524 and eluted at 5.30 min (Figure 4). The accurate mass measurements predicated that the chemical formula was C_43_H_62_O_10_N_7_ (mass error < 3.5 ppm), indicating that the peptide bond between leucine and alanine of seongsanamide A was cleaved. The ions retention time at 6.39, 6.69, 6.89, and 7.77 min in the *m/z* 836.5 channel was the in-source dissociation product of M4, M3, M2, and seongsanamide A by the peptide bond cleavage between each leucine and alanine, respectively. In-source dissociation in mass spectrometry is very common for drug conjugates, such as sulfate [40], glucuronide [41,42], acetyl [41], and glutathione [43] conjugates. Liu et al. (2010) identified the metabolite structure of actinomycins, which also has a cyclodepsipeptide ring structure, using an in-source collision-induced dissociation technique [44]. In-source collision-induced dissociation was also applied to the characterization of microcystins, a type of cyclic peptides, from a *Microcystis aeruginosa culture* [45]. The metabolite M1 produced fragment ions at *m/z* 808.4604 and 765.4148 through the loss of a carbon monoxide and an amide bond cleavage from the protonated molecular ion, respectively (Figure 5B). Fragment ions at *m/z* 737.4225 and 723.3703 were generated through the loss of a carbon monoxide and the loss of isopropyl moiety from fragment ion at *m/z* 765.4148, respectively (Appendix A). The metabolites M2, M3, and M4 exhibited a protonated molecular ion (M + H)^+^ at *m/z* 1021.5589, 1037.5519, and 1053.5475, which were 16, 32, and 48 Da higher than seongsanamide A, indicating mono-, di-, and tri-hydroxylation of seongsanamide A, respectively. They eluted at 6.88, 6.70, and 6.40 min (Figure 4). The metabolite M2 produced fragment ions at *m/z* 1003.5486 through the loss of a water molecule from the protonated molecular ion (Figure 5C). Fragment ions at *m/z* 975.5534 and 836.4548 were generated through the loss of a carbon monoxide and amide bond cleavage from the fragment ion at *m/z* 1003.5486, respectively (Figure 5C). Fragment ions at *m/z* 808.4587, 765.4170, and 737.4220 were generated through the loss of carbon monoxide from the base peak, the loss of an alanine residue (−71 Da) from the base peak, and the loss of a carbon monoxide from the *m/z* 765.4170 peak, respectively (Appendix A). The metabolite M3 yielded a characteristic fragment ion at *m/z* 1001.5328 and 836.4549, indicating the loss of two water molecules and the peptide bond cleavage between the leucine and alanine of seongsanamide A, respectively (Figure 4 and Figure 5D); and fragment ions at *m/z* 808.4589 and 765.4170 through the loss of a carbon monoxide and the loss of an alanine residue (−71 Da) from the base peak, respectively (Appendix A). Metabolite M4 produced a characteristic fragment ion at *m/z* 999.5174 and 836.4543, indicating the loss of three water molecules and the peptide bond cleavage between the leucine and alanine of seongsanamide A, respectively (Figure 4 and Figure 5E). Fragment ions at *m/z* 808.4595 and 765.4170 were generated through the loss of a carbon monoxide and the loss of an alanine residue (−71 Da) from the base peak, respectively (Appendix A).

### 3.4. Characterization of Human Cytochrome P450 Enzymes Responsible for the Formation of Four Metabolites of Seongsanamide A

Incubation of seongsanamide A in the presence of NADPH in HLMs resulted in the formation of mono- and di-hydroxyseongsanamide A (M2 and M3) as major metabolites. M1 and M4 were detected as minor metabolites. The formation of four metabolites from seongsanamide A was examined using eleven human rP450 isoforms. We found that CYP3A4 played a major role in the formation of M1, M2, M3, and M4, whereas CYP3A5 was involved in the formation of mono-hydroxyseongsanamide A (M2) (Figure 6). CYP3A is known as an enzyme primarily involved in the hydroxylation of cyclosporine [46], a cyclic peptide, and saquinavir [46,47], a peptide derivative.

### 3.5. In Vitro Inhibition of Cytochrome P450 Enzymes by Seongsanamide A

Inhibition of P450 activity was evaluated at seongsanamide A concentrations of up to 50 µM to determine the effect on P450-mediated drug interactions in HLMs. Seongsanamide A inhibited CYP3A-mediated midazolam 1′-hydroxylation with an IC_50_ value of 13.1 µM, whereas it inhibited CYP2B6, CYP2C19, and CYP2D6 activities with IC_50_ values ranging from 21.9 to 25.5 µM (Table 2). However, the inhibitory potential of seongsanamide A against CYP3A was much lower than that of cyclosporine (IC_50_ = 1.24 µM) [11] and saquinavir (IC_50_ = 2.14 µM) [48] which are known peptide drugs. Seongsanamide A at a concentration of 50 µM did not affect the activities of the other eight P450 isoforms. These findings suggest that drug interactions between seongsanamide A and these P450s would not be expected to interfere with the metabolism of other drugs.

In conclusion, our results indicate that both metabolomic and FBMN approaches are efficient tools for identifying drug metabolites in HLM incubation samples. One hydrolyzed (M1) and three hydroxylated metabolites (M2, M3, and M4) were identified based on accurate mass and product ion scan mass spectra using LC-HRMS. Mono- (M2) and di-hydroxyseongsanamide A (M3) are the most abundant metabolites, and CYP3A is the major metabolizing enzyme involved in the enzymatic hydrolysis and hydroxylation of seongsanamide A. In addition, seongsanamide A was found to have weak inhibitory effects on nine cytochrome P450 enzymes (IC_50_ > 10 µM). These data will be useful for understanding drug metabolism, pharmacokinetics, and drug interactions of seongsanamide A in vivo. This study demonstrated that the untargeted metabolomics and FBMN methods could be promising tools for the identification of drug metabolites.

## Figures and Tables

**Figure 1 pharmaceutics-13-01031-f001:**
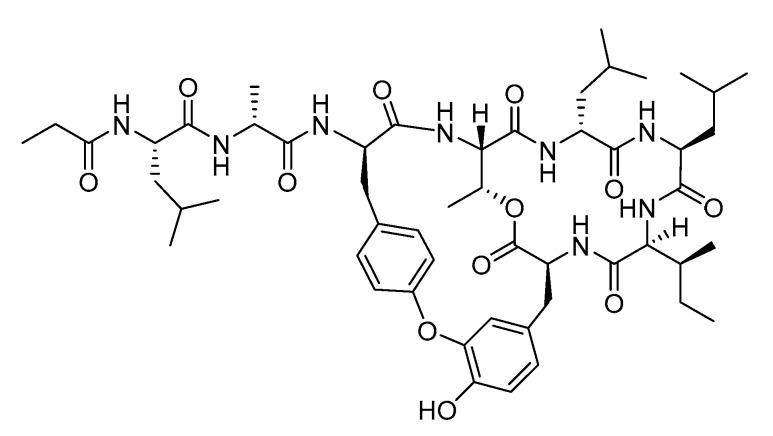
Chemical structure of seongsanamide A.

**Figure 2 pharmaceutics-13-01031-f002:**
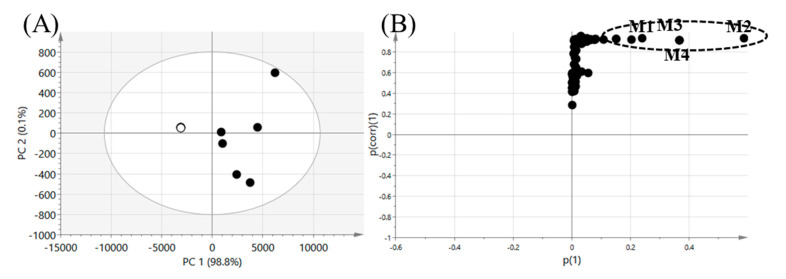
Multivariate analysis of seongsanamide A metabolites in human liver microsomes (HLMs). (**A**) Scores scatter plot generated by a principal component analysis (PCA) from the heat-deactivated HLMs (○) and normal HLMs (●). (**B**) Loading S-plot generated by an orthogonal partial least-squares-discriminant analysis (OPLS-DA) model. Identified metabolites (M1–M4) are marked with abbreviations on the S-plot, and metabolite lists are provided in Table 1.

**Figure 3 pharmaceutics-13-01031-f003:**
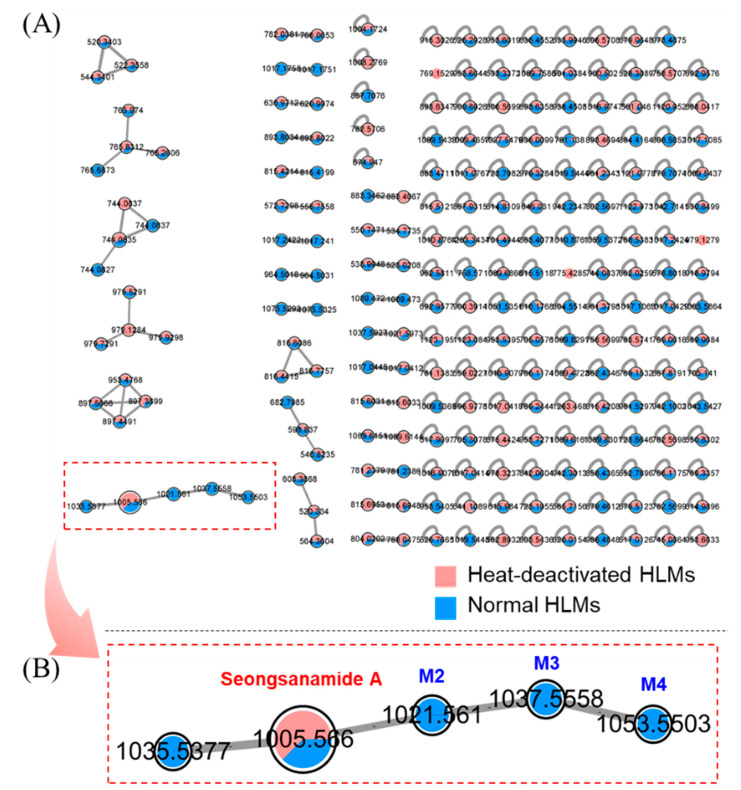
Visualization of in vitro seongsanamide A metabolism using feature-based molecular networking. Seongsanamide A was incubated with human liver microsomes (HLMs). (**A**) The overall multi-matrix molecular network. (**B**) Details of the specific seongsanamide A containing cluster. Nodes are labeled with the *m/z* value and identified metabolite name (M2, mono-hydroxyseongsanamide A; M3, di-hydroxyseongsanamide A; and M4, tri-hydroxyseongsanamide A). Red nodes represent the ions from the heat-deactivated HLMs, and blue nodes represent the ions from the normal HLMs.

**Figure 4 pharmaceutics-13-01031-f004:**
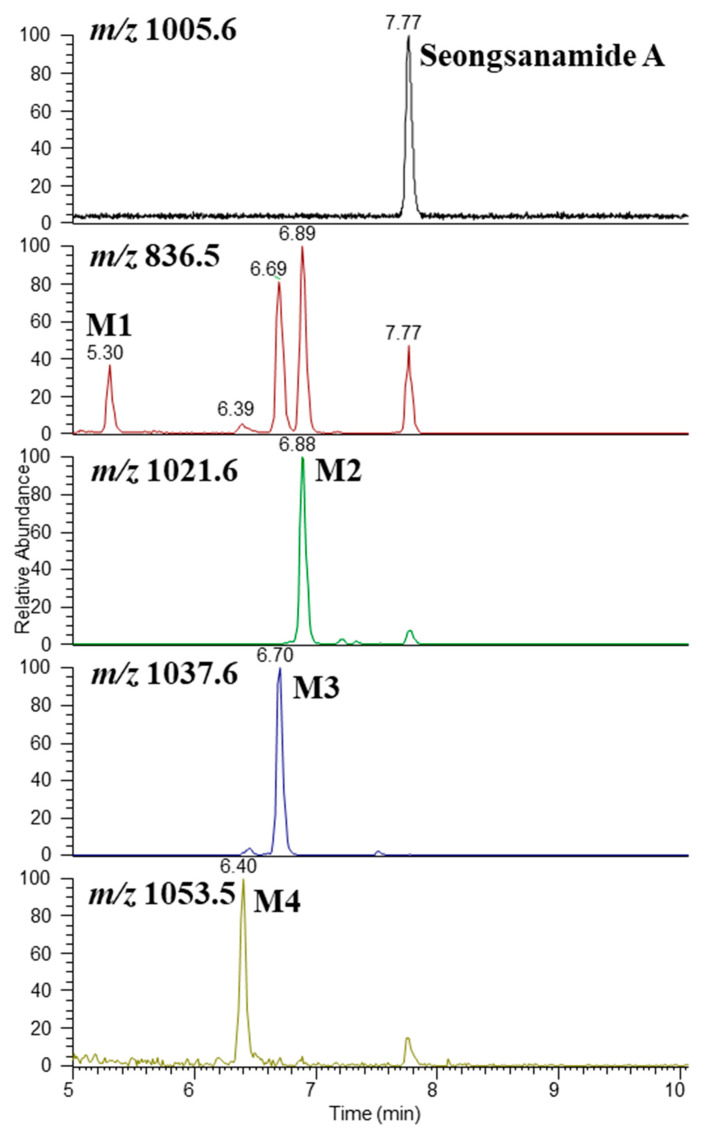
Representative selected-ion monitoring chromatogram of seongsanamide A and its four metabolites (M1–M4) obtained from the liquid chromatography-high resolution mass spectrometry analysis of human liver microsomes.

**Figure 5 pharmaceutics-13-01031-f005:**
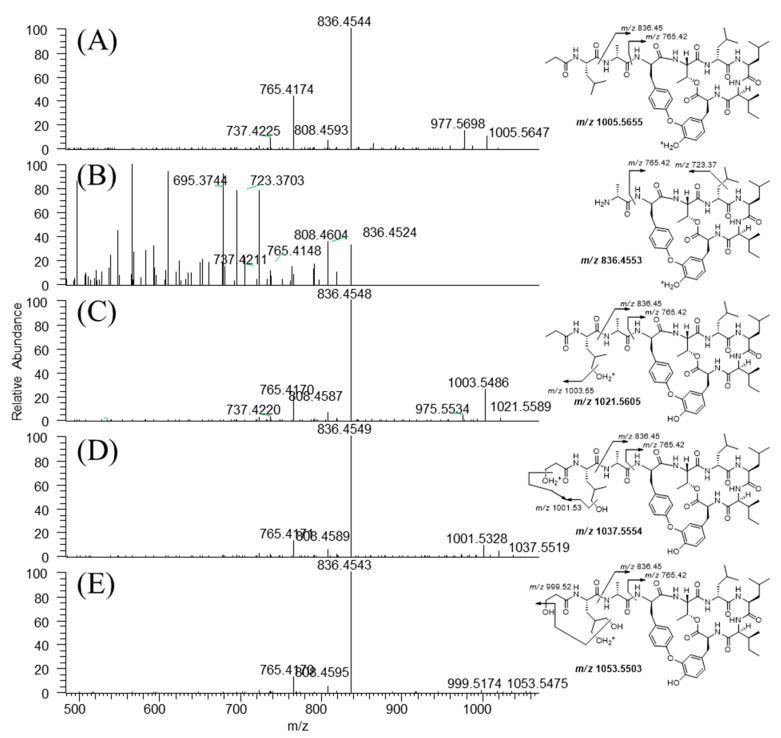
Representative product ion scan mass spectra and proposed fragmentation patterns of seongsanamide A (**A**), hydrolyzed seongsanamide A (M1), (**B**), Mono- (M2), (**C**), Di- (M3), (**D**), and tri-hydroxyseongsanamide A (M4), (**E**).

**Figure 6 pharmaceutics-13-01031-f006:**
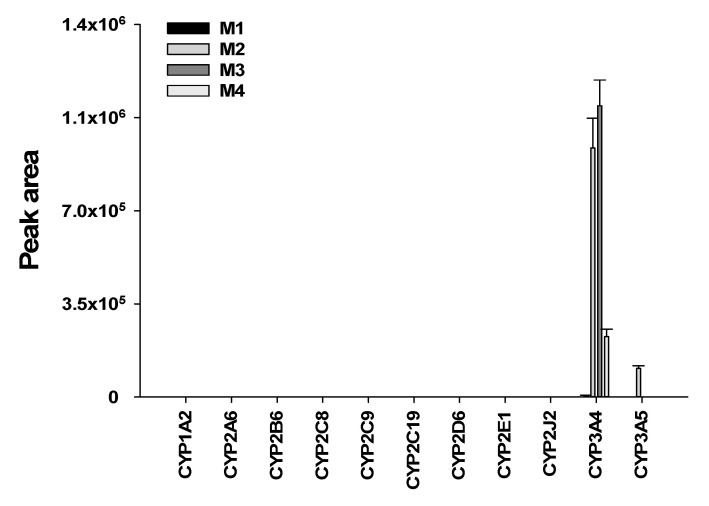
Representative plots of the formation of hydrolyzed seongsanamide A (M1), mono- (M2), di- (M3), and tri-hyroxyseongsanamide A (M4) from seongsanamide A by recombinant human cytochrome P450 isoforms. Human recombinant cytochrome P450s 1A2, 2A6, 2B6, 2C8, 2C9, 2C19, 2D6, 2E1, 2J2, 3A4, and 3A5 were incubated with seongsanamide A (1 µM) at 37 °C for 15 min in the presence of an NADPH generating system. Error bars represent standard deviations (*n* = 3).

**Table 1 pharmaceutics-13-01031-t001:** Summary of the mass spectral data of seongsanamide A and its four metabolites identified in the human liver microsomal incubation sample.

No.	Metabolic Pathway	t_R_ (min)	*m/z* ((M + H)^+^)		Error (ppm)	Formula (Neutral)
			Measured	Theoretical		
P	Seongsanamide A	7.8	1005.5647	1005.5655	–0.822	C_52_H_77_O_12_N_8_
M1	Hydrolysis	5.3	836.4524	836.4553	–3.368	C_43_H_62_O_10_N_7_
M2	Mono-hydroxylation	6.9	1021.5589	1021.5605	–1.459	C_52_H_77_O_13_N_8_
M3	Di-hydroxylation	6.7	1037.5519	1037.5554	–3.263	C_52_H_77_O_14_N_8_
M4	Tri-hydroxylation	6.4	1053.5475	1053.5503	–2.639	C_52_H_77_O_15_N_8_

**Table 2 pharmaceutics-13-01031-t002:** Inhibitory potential of seongsanamide A against the activities of nine cytochrome P450 isoforms in human liver microsomes.

Activity	P450 Isoforms	IC_50_ (µM)
Phenacetin *O*-deethylation	CYP1A2	>50
Coumarin 7-hydroxylation	CYP2A6	>50
Bupropion hydroxylation	CYP2B6	21.9
Amodiaquine *N*-deethylation	CYP2C8	>50
Tolbutamide 4-methylhydroxylation	CYP2C9	>50
Omeprazole 5-Hydroxylation	CYP2C19	25.5
Dextromethorphan *O*-demethylation	CYP2D6	23.7
Chlorzoxazone 6-hydroxylation	CYP2E1	>50
Midazolam 1′-hydroxylation	CYP3A	13.1

## Data Availability

All data in this study have been included in this manuscript.

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
