# Peer review of "In Vitro Metabolism Study of Seongsanamide A in Human Liver Microsomes Using Non-Targeted Metabolomics and Feature-Based Molecular Networking"

_pharmaceutics, 2021, doi:10.3390/pharmaceutics13071031_

Round 1
Reviewer 1 Report
Aim of this study is to elucidate in vitro metabolic pathway associated with seongsanamide A. The authors demonstrated that combined metabolomics and FBMN approaches contributed to identify drug metabolites in HLM incubation samples. This work may be technically sound, but I feel that the method description in the main text can be improved. From the viewpoint of the open science and transparency, you should pay attention to share your metabolome dataset, including metabolite identification/annotation. My specific comments and suggestions are as follows.
Specific comments:
(1) Methods
There was no description about parameter settings of software tools in this study. For example, the Compound Discover. Detailed description of data pre-treatment/transformation is also lacking. For example, PCA and OPLS-DA. Did you use log-transform/auto-scaling in the multivariate statistical analyses?
(2) Figures and Tables
Figure 2: The font size in x-y axes is too small!
Figure 3: Why are there nodes with self-loop? For example, the m/z value name “1004.1724.”
(3) Data Availability Statement
You should open all the metabolome data in a public repository, like MetaboLights and Metabolomics Workbench. I think that this contributes to enhance research transparency and reproducibility.
(4) Discussion and conclusion
The section Results and Discussion is quite poor. Your conclusion too. For example, you should include your future perspectives in conclusion.
Minor comments/suggestions/errors:
(1) The terms “metabolomics” and “FBMN” listed in the title. In Keywords, the authors should reconsider including another keywords representing this work.
(2) Would you please share the cytoscape related files related to the overall multi-matrix molecular network?
(3) The whole manuscript needs to be revised by native speaker.
Reviewer 2 Report
Attached.

Author Response
Please see the attachement.

Reviewer 3 Report
The study investigated metabolism and drug interactions of seongsanamide A in vitro in human liver microsomes by using non-targeted metabolomics
and feature-based molecular networking (FBMN) techniques. All methods are well described with some minor lacks of clarity.
Major comments:
I would recommend addition of some citations on metabolism of peptide drugs in the introduction, for instance cyclosporine is a prominent cyclic peptide drug that is metabolized by CYPs.
It would be good, if you can motivate why you propose these positions of the OH-groups in the structures of the hydroxylated metabolites M2-M4. Also other positions in the seongsanamide A could be oxidized by CYPs.
Minor comments:
Lines 59-61: Please add the correct incubation time used for the investigation of the metabolites. There are 2 different values (30 and 60 min).
Lines 133-135: How long have you incubated the HLMs in the presence or absence of seongsanamide A? 0min is stated. In my opinion , it would be also better to add the concentrations of seongsanamide A tested for inhibition of CYP activities instead of the concentration range.
